# FedIN: Federated Intermediate Layers Learning for Model Heterogeneity

## Abstract

Federated learning (FL) facilitates edge devices to cooperatively train a global shared model while maintaining the training data locally and privately. However, a prevalent yet impractical assumption in FL requires the participating edge devices to train on an identical global model architecture. Recent research endeavors to address this problem in FL using public datasets. Nevertheless, acquiring data distributions that closely match to those of participating users poses a significant challenge. In this study, we propose an FL method called Federated Intermediate Layers Learning (FedIN), which supports heterogeneous models without relying on any public datasets. Instead, FedIN leverages the inherent knowledge embedded in client model features to facilitate knowledge exchange. To harness the knowledge from client features, we propose Intermediate Layers (IN) training to align intermediate layers based on features obtained from other clients. IN training only needs minimal memory and communication overhead by employing a single batch of client features. Additionally, we formulate and resolve a convex optimization problem to mitigate the challenge of gradient divergence stemming from model heterogeneity. The experimental results demonstrate the superior performance of FedIN in heterogeneous model settings compared to state-of-the-art algorithms. Furthermore, the experiments discuss the details of how to protect user privacy leaked from IN features, and our ablation study illustrates the effectiveness of IN training.

## 1 Introduction

The substantial surge in Internet-of-Things (IoT) device utilization has led to the generation of vast quantities of user data (Song et al., 2022). Effectively managing this IoT big data without compromising user privacy has emerged as a significant concern. **Federated Learning** (FL) (McMahan et al., 2017) is proposed as a distributed machine learning paradigm that facilitates collaborative training on IoT data while keeping user data locally. Within FL, each client transmits model weights from their local models to the server following a few local training epochs. Subsequently, the server aggregates these weights to update the global model and sends this model back to clients.

While Federated Learning (FL) has demonstrated success in various applications, such as recognizing human activities (Chen et al., 2019b; Ouyang et al., 2021) and learning sentiment (Smith et al., 2017; Qin et al., 2021), numerous practical challenges persist within the FL domain (Kairouz et al., 2021). One of the most crucial and practical challenges is system heterogeneity, characterized by varying resources among client devices participating in FL training (Li et al., 2020a; Chan et al., 2024). Many existing FL schemes (Li et al., 2021a; Karimireddy et al., 2020) assume that the client devices with distinct resources possess the same architecture as the global shared model for global aggregation. Nevertheless, clients with limited computation resources may struggle to complete local training in time, dragging the training speed of the entire communication round. The clients hindering the training process are called stragglers. To combat this issue, some research has proposed asynchronous FL (Xie et al., 2020; Chen et al., 2020; Chai et al., 2021), adjusting local training epochs dynamically and clustering clients according to their available resources to mitigate the problem of stragglers. Nevertheless, given that all clients keep the same model architecture, less capable clients may lack sufficient memory to deploy the shared global model. In this case, the global model must be adjusted to a smaller size, leading to a resource waste of more capable clients and diminishing the

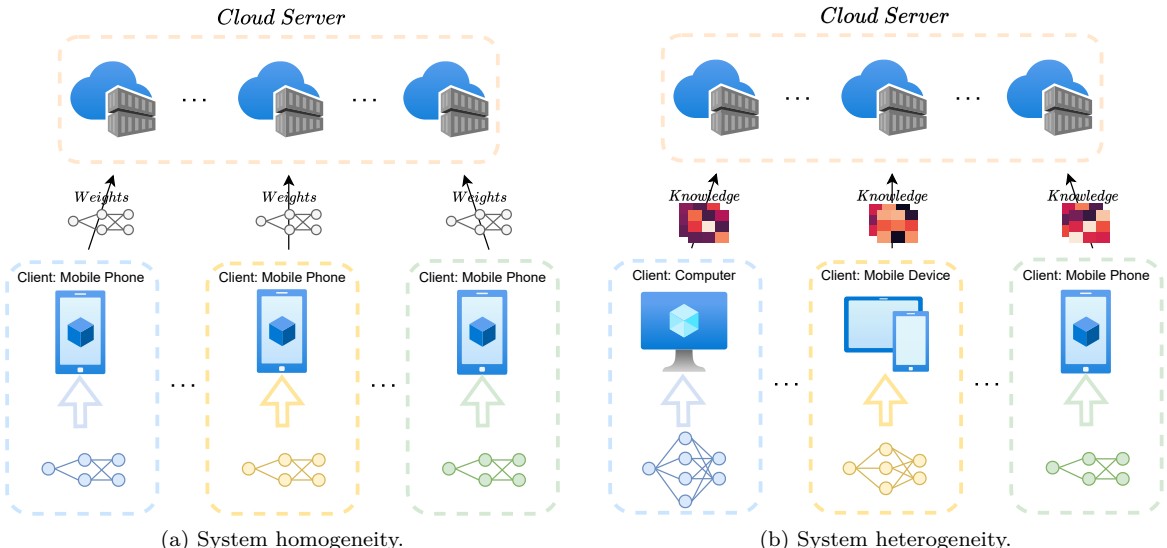

(a) System homogeneity.  (b) System heterogeneity.

Figure 1: All clients have the same model architectures in system homogeneous FL as shown in Figure 1a. In system heterogeneity, the clients participate in the federated learning with different available resources, inducing different model architectures in Figure 1b.

performance of FL training. Therefore, A straightforward way to facilitate system heterogeneity is to deploy different model architectures based on the available resources of the clients, as shown in Figure 1b. However, the server can not aggregate the weights directly like Figure 1a under heterogeneous model architectures. Recent works addressing this challenge through knowledge distillation (Hinton et al., 2015) using a public dataset, such as RHFL (Fang & Ye, 2022) and FedMD (Li & Wang, 2019). While these methods allow for diverse model architectures on clients, it is challenging to collect a suitable public dataset with a similar distribution to the local datasets.

Therefore, to support system heterogeneity without relying on a public dataset, we propose a method called Federated Intermediate Layers Learning (**FedIN**), training the intermediate layers according to a single batch of features obtained from other clients. In FedIN, a local model architecture consists of three components: an extractor, intermediate layers, and a classifier, as depicted in Figure 2. Client features are derived from the outputs of the extractor and the inputs to the classifier. Notably, clients only need to transmit **one batch** of features to the server, in addition to weight updates. The intermediate layers are updated through a combination of local training and **IN** training process, where IN training leverages a single batch of features to extract latent knowledge from other clients. However, directly deploying these two training processes can induce a critical problem called gradient divergence (Wang et al., 2020; Zhao et al., 2018), as the latent information from the local dataset and the features collected from other clients varies, particularly in a model heterogeneous environment. To alleviate the effect of this problem, we formulate and address a convex optimization problem to obtain the optimal updated gradients. Moreover, we use a simple yet efficient method, adding Gaussian noise to the client features to protect user privacy. The experiment results reveal that FedIN outperforms the baselines in terms of both accuracy and overhead.

Our contributions are summarized as follows.

- We proposed a novel FL method called **FedIN**, utilizing local training and IN training for intermediate layers, which is a flexible and reliable FL method addressing the system heterogeneity problem.
- To alleviate the effects of the gradient divergence, we formulate a convex optimization problem to derive the optimal updated gradient. The ablation study shows its effectiveness in handling the gradient divergence problem.
- To protect user privacy within FedIN, we utilize Gaussian noise in the IN training process. The experiments demonstrate the effectiveness of this approach in ensuring user privacy.
- Our experiments reveal that FedIN achieves the best performances in the IID and non-IID data compared with the state-of-the-art algorithms. Moreover, we conduct a thorough analysis to investigate the factors contributing to the improvements attained by FedIN.

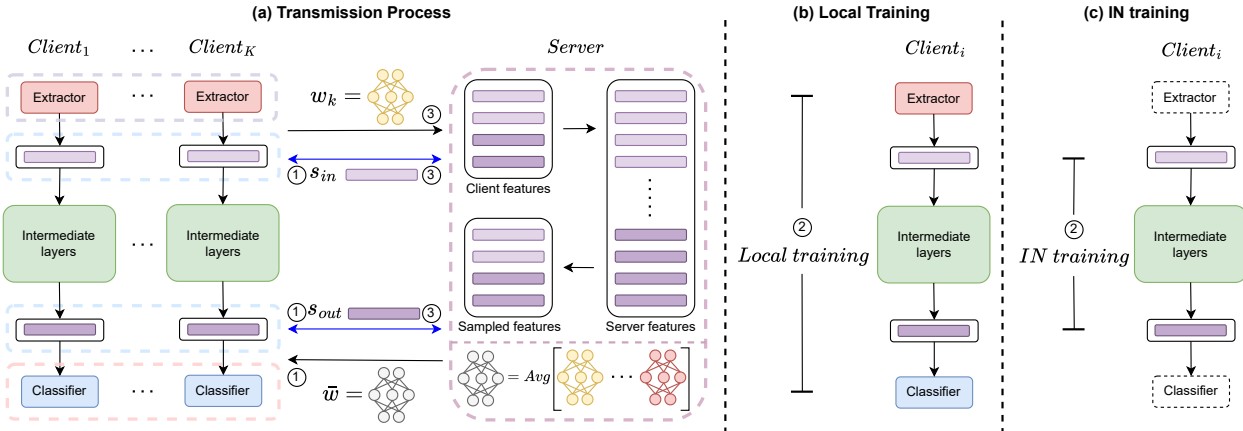

Figure 2: **Details of (a) the transmission process, (b) local training, and (c) IN training process for FedIN.** The process for FedIN is described as follows. ① First, clients receive client features and global weights $\bar{w}$ from the server. ② After updating client weights by global weights, the clients complete the local training from the local private dataset and complete the IN training for the client features inputs and outputs $(s_{in}, s_{out})$ from the server. ③ Upon completing the training process, clients transmit the model weights and new client features, denoted as $(w_k, s_{in}, s_{out})$, to the server. The aggregation methods for system heterogeneity are discussed in section 4.4.

## 2 Related Work

### 2.1 Federated Learning

Federated Learning (FL) was proposed in 2017 to organize cooperative model training among edge devices and servers (McMahan et al., 2017). In FL, numerous clients train models jointly while retaining training data locally to maintain privacy protection. Various methods have been proposed and achieved good performance in different scenarios. In (Xie et al., 2020), FedAsyn utilizes coordinators and schedulers to create an asynchronous training process, handling the stragglers in the FL training process. FedProx (Li et al., 2020b) regularizes and re-parametrizes FedAvg, guaranteeing convergence when learning over non-IID data. To share local knowledge among clients with different model architectures, FCCL (Huang et al., 2022) generates a cross-correlation matrix based on the unlabeled public dataset.

### 2.2 Heterogeneous Models

Our work focuses on supporting heterogeneous models in FL. This subsection classifies recent research contributing to model heterogeneity into three categories.

**Public and Auxiliary Data.** If a server has a public dataset, clients can exploit the general knowledge from this dataset, constructing a simple and efficient bridge to exchange knowledge among clients. FedAUX (Sattler et al., 2021) utilizes unsupervised pre-training and unlabeled auxiliary data to initialize heterogeneous models. FedGen (Zhu et al., 2021) simulates the prior knowledge from all the clients according to a generator. To dig out the latent knowledge from the public dataset, several studies (Li & Wang, 2019; Li et al., 2021b; He et al., 2020) propose addressing the system heterogeneity problem, inspired by knowledge distillation (Hinton et al., 2015). In FedMD (Li & Wang, 2019), a large public dataset is deployed in a server, while the clients distill and transmit logits from this dataset to learn the knowledge from both logits and local private datasets. In FedH2L (Li et al., 2021b), clients extract the logits from a public dataset consisting of small portions of local datasets from other clients. In RHFL (Fang & Ye, 2022), a server calculates the weights of clients by the symmetric cross-entropy loss function, and clients distilled knowledge from the unlabeled dataset. FCCL (Huang et al., 2022) computed a cross-correlation matrix also based on the unlabeled public dataset. MocoSFL (Li et al., 2023) proposes a mechanism, replay memory on features to assist the MoCo functions (Chen et al., 2021), a contrastive framework, in model heterogeneous FL.

**Data-free Knowledge Distillation.** The basic ideas of data-free KD are to optimize noise inputs to minimize the distance to prior knowledge (Nayak et al., 2019), and Chen et al. (Chen et al., 2019a) train Generative Adversarial Networks (GANs) (Goodfellow et al., 2014) to generate training data for the entire KD process, utilizing the knowledge distilled from the teacher model. To free the limitation from a public dataset, some research works consider data-free KD in FL. In FedML (Shen et al., 2020), latent knowledge from homogeneous models is applied to train heterogeneous models. In FedHe (Chan & Ngai, 2021), logits belonging to the same class are directly averaged in a server. In FedGKT (He et al., 2020), a neural network is split into a client and a server, while the server completes the entire training process based on the features and logits collected from all clients. FedMK (Liu et al., 2023) utilizes dataset distillation to transmit latent knowledge between clients in FL.

**Splitting Models.** To adapt to the available resources of different clients, several studies split the large models into small sub-models. HeteroFL (Diao et al., 2021) divides a large model into local models with different sizes. However, the architectures of local and global models are still restricted by the same model architecture. SlimFL (Baek et al., 2022) integrates slimmable neural network (SNN) architectures (Yu & Huang, 2019) into FL, adapting the widths of local neural networks based on resource limitations. In (Horvath et al., 2021), FjORD leverages Ordered Dropout and a self-distillation method to determine the model widths. ScaleFL (Ilhan et al., 2023) splits a server model along two dimensions, and local models are trained using the cross-entropy and KL-divergence loss functions. InCo (Chan et al., 2024) proposes three splitting methods with convex optimization problems to solve the gradient divergence problem in heterogeneous FL.

## 3 Problem Formulation

The goal of FL is to collaborate with the clients to train a shared global model while keeping their local data private. We briefly summarize the optimization problem below. We assume that $K$ clients participate in FL. Each client has a private dataset $D_k = \{(x_{i,k}, y_{i,k}) | i = 1, 2, ..., |D_k|\}$, where $k \in \{1, ..., K\}$ is the index of a client, and $|D_k|$ denotes the size of a dataset $D_k$. Private dataset $D_k$ is only accessible to client $k$, guaranteeing data privacy. In traditional FL, the clients share identical model architecture. We denote a training model by $f(x; w)$, where $w$ are the training weights and $x$ are the inputs. The loss function $l_k$ of client $k$ is shown as follows,

$$\min_{w} \quad l_k(w) = \frac{1}{|D_k|} \sum_{i=1}^{|D_k|} l(f(x_{i,k}; w), y_{i,k}), \tag{1}$$

where $l(\cdot, \cdot)$ is a loss function for each data sample $(x_{i,k}, y_{i,k})$. Nevertheless, it may not be possible to deploy an identical model architecture for all the clients due to system heterogeneity. One potential solution is to allow clients to select different model architectures according to their capabilities in heterogeneous FL. The problem of heterogeneous FL is described as follows. We denote $w_k$ as the model weights of client $k$. If the total size of all datasets is $N = \sum_{k=1}^{K} |D_k|$, the global optimization function is described as follows,

$$\min_{w_1, w_2, ..., w_K} \quad L(w_1, ..., w_K) = \sum_{k=1}^{K} \frac{|D_k|}{N} l_k(w_k), \tag{2}$$

where the optimized model weights $\{w_1, w_2, ..., w_K\}$ have different sizes. Thus, the direct aggregation of entire model weights becomes unfeasible when dealing with heterogeneity among models. Therefore, we adopt layer-wise heterogeneous aggregation (Liu et al., 2022; Chan et al., 2024) as an alternative approach to aggregating the layer weights of heterogeneous models instead of the entire model weights in our experiments.

## 4 FedIN: Federated Intermediate Layers Learning

In this section, we describe the details of FedIN, focusing on addressing system heterogeneity by deploying clients with diverse model architectures that align with their available resources. Figure 2 illustrates the

workflow of FedIN. The client model consists of three key components: an extractor, intermediate layers, and a classifier. The outputs of the extractor, referred to as feature inputs ($s_{in}$), serve as inputs to the intermediate layers. Similarly, the outputs of the intermediate layers, referred to as feature outputs ($s_{out}$), act as inputs to the classifier. The client features are the pair of feature inputs and outputs, denoted as ($s_{in}, s_{out}$). To be specific, FedIN encompasses two training processes: local training, which leverages the private dataset, and IN training, which relies on the feature inputs and outputs ($s_{in}, s_{out}$). Moreover, to address the challenge of gradient divergence arising from conflicts from model heterogeneity, we propose a convex optimization problem formulation to obtain the optimal updated gradients.

## 4.1 Local Training and IN Training

The clients receive a single batch of feature inputs and feature outputs, denoted as $S = \{(s_{i,in}^c, s_{i,out}^c)|i = 1, 2, ..., |S|\}$, from the server. These samples are utilized for training the intermediate layers during the IN training process. The superscript $c$ means that these feature inputs and outputs are from the central server. The clients begin their local training after receiving a batch of client features from the server. For an instance $(x_{i,k}, y_{i,k}) \in D_k$, client $k$ conducts local training on its private dataset. The loss function of the local training is shown as follows,

$$l_{local,k} = \ l_{CE}(f(x_{i,k}; w_k^t), y_{i,k}) + \frac{\mu}{2}||w_k^t - w_k^{t-1}||^2, \tag{3}$$

where $w_k^t$ are the weights of client $k$ at time $t$, and $l_{CE}$ is the cross-entropy loss function for the local training. To ensure client consistency, we add a proximal regularization term (Li et al., 2020b) in Eq. 3.

The second training process is IN training, which is training the intermediate layers from the features dataset $S$. It is worth mentioning that the sample number of $S$ is one batch size. We denote the weights of the extractor and the classifier by $w_{e,k}$ and $w_{c,k}$ for client $k \in \{1, ..., K\}$. Moreover, the weights of the intermediate layers are denoted by $w_{in,k}$. The relations among the data sample $(x_{i,k}, y_{i,k}) \in D_k$, client weights, and $(s_{i,in}^k, s_{i,out}^k)$ are shown as follows,

$$s_{i,in}^k = f(x_{i,k}; w_{e,k}), \tag{4}$$

$$s_{i,out}^k = f(s_{i,in}^k; w_{in,k}), \tag{5}$$

$$f(x_{i,k}; w_k) = f(s_{i,out}^k; w_{c,k}). \tag{6}$$

Eq. 4 shows that the feature input $s_{i,in}^k$, the light purple components in Figure 2 is the output of the extractor $w_{e,k}$ of an instance $(x_{i,k}, y_{i,k})$ from client $k$. Eq. 5 describes that the feature output $s_{i,out}^k$, the deep purple components in Figure 2 is the output of the intermediate layers $w_{in,k}$ with the feature input $s_{i,in}^k$. Eq. 6 proves the equivalence between the output of the classifier $w_{c,k}$ and the output of the whole client model $w_k$. Eq. 5 shows the main function of the IN training, as shown in Figure 2(c). After the client receives the feature dataset $S = \{(s_{i,in}^c, s_{i,out}^c)|i = 1, 2, ..., |S|\}$, it begins the IN training for the intermediate layers. The feature inputs $s_{i,in}^c$ from the server are the inputs of the intermediate layers, while the $s_{i,out}^c$ are the targets of the IN training. The loss function of IN training is defined as follows,

$$l_{IN,k} = l_{MSE}(f(s_{i,in}^c; w_{in,k}), s_{i,out}^c), \tag{7}$$

where $l_{MSE}$ is a mean-square error loss function. The weights $w_{in,k}$ are updated by the loss functions of the local training $l_{local,k}$ and the IN training $l_{IN,k}$. We use MSE as the loss function due to its effectiveness in this learning method. Moreover, $s_{in}$ and $s_{out}$ do not represent probability distributions, making it difficult to incorporate other losses such as KL divergence and cross-entropy losses.

## 4.2 Gradient Alleviation

However, local training is based on the local data, while IN training is based on the features from other clients' data. Different local datasets lead to varied distributions, resulting in dissimilar optimized directions. Moreover, in our scenario, deploying distinct model architectures in clients emphasizes differences in feature spaces, as shown in Figure 3. These combined factors result in di-

vergent gradients between local training and IN training, impeding the pace of convergence and disturbing the model to achieve the optimum point (Wang et al., 2020; Zhao et al., 2018). Therefore, mitigating this gradient divergence is imperative for the effectiveness of our method. To address this problem, inspired by (Chan et al., 2024), we formulate a convex optimization problem as follows.

We define the gradients from the local training as a matrix $G_{local}$ and the gradients from the IN training as a matrix $G_{IN}$, both for the intermediate layers. To guarantee the optimized direction of the models, we design a constraint for the gradient as follows,

$$\langle G_{IN}, G_{local} \rangle \geq 0, \tag{8}$$

where $\langle \cdot, \cdot \rangle$ is the dot product, which ensures the optimized direction for $G_{local}$ and $G_{IN}$ to be the same. In the optimization problem, we denote the new optimized gradients by a matrix $Z$ and model the following convex optimization primal problem,

$$\min_{Z} ||G_{IN} - Z||_F^2, \quad s.t. \ \langle Z, G_{local} \rangle \geq 0, \tag{9}$$

where we maintain the optimized direction between $Z$ and $G_{local}$ to be the same and minimize the distance between $Z$ and $G_{in}$. We consider that the information from the feature inputs and outputs is

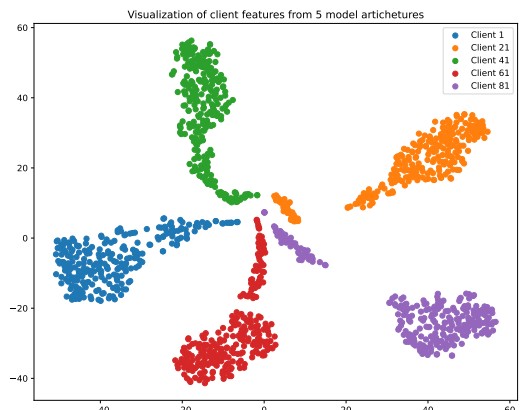

Figure 3: T-SNE visualization depicts IN feature outputs $s_{out}$ derived from five distinct model architectures with each color representing a unique model architecture.

more fruitful than the local private dataset which is easier to have over-fitting in the training process. We solve this convex optimization problem by the Lagrange dual problem (Bot et al., 2009; Chan et al., 2024). The Lagrangian is shown as,

$$L(Z, \lambda) = tr(G_{IN}^T G_{IN}) - tr(Z^T G_{IN})$$
$$-tr(G_{IN}^T Z) + tr(Z^T Z) - \lambda tr(G_{local}^T Z), \tag{10}$$

where $tr(A)$ means the trace of the matrix $A$, and the $\lambda$ is a Lagrange multiplier associated with $\langle Z, G_{local} \rangle \geq 0$. To derive the dual problem, we first get the optimum of $Z$ for the Lagrangian Eq. 10, and then obtain the Lagrange dual function $g(\lambda) = \inf_Z L(Z, \lambda)$. Thus, the Lagrange dual problem is described as follows,

$$\max_{\lambda} \ g(\lambda) = -\frac{\lambda^2}{4} tr(G_{local}^T G_{local}) - \lambda tr(G_{local}^T G_{IN}), \quad s.t. \ \lambda \geq 0, \tag{11}$$

where the optimum of the Lagrangian Eq. 10 is $Z = G_{IN} + \frac{\lambda}{2} G_{local}$. If the $\lambda$ is large enough, it is obvious that $\langle Z, G_{local} \rangle > 0$, which means this convex optimization problem holds strong duality because it satisfies the Slater's constraint qualification(Boyd et al., 2004), i.e., the optimum of the primal problem Eq. 9 is also $Z = G_{IN} + \frac{\lambda}{2} G_{local}$. Furthermore, the dual problem Eq. 11 can be solved to obtain the analytic solution for $\lambda$ and $Z$, which is shown as follows,

$$Z = \begin{cases} G_{IN}, & \text{if } b \geq 0 \\ G_{IN} - \frac{b}{a} G_{local}, & \text{if } b < 0 \end{cases} \tag{12}$$

where $a = tr(G_{local}^T G_{local})$ and $b = tr(G_{local}^T G_{IN})$. However, one crucial point is that the clients will handle this optimization process. If we calculate each gradient matrix following Eq. 12, this process would occupy lots of computing resources because of the matrix multiplication. Therefore, to mitigate the computational pressure on the clients, we simplified the updated gradient matrix as,

$$Z = G_{IN} + \frac{\lambda}{2} G_{local}, \tag{13}$$

where $\lambda = 1$ is set for the optimum point of the primal problem in our experiment settings. The detailed derivation process for this section is shown in Appendix A.

### 4.3 Privacy Consideration

In our methods, clients are required to transmit feature inputs and outputs to the server, raising privacy concerns regarding the potential leakage of private data through transmitted features. We investigate two recent related attack methods, the Gradient Inversion Attack and the Model Inversion Attack. The Gradient Inversion Attack relies on the strong assumption that the server knows the private statistic of BatchNorm (Huang et al., 2021), which is not appropriate to FedIN as such information is unnecessary to transmit to the server.

Additionally, the Model Inversion Attack poses a greater risk of stealing private information in our scenario, but one strong assumption for this attack is that the server needs to have prior knowledge of the client input images (Li et al., 2022), which is impractical in our scenario as the server does not receive any images from the clients. However, as the server accesses the model parameters and the IN feature inputs and outputs, we explore an alternative method known as dataset distillation (Wang et al., 2018; Lei & Tao, 2023) to potentially reconstruct the private images from the clients. We randomly initialize and train a batch of noise $\hat{x}$ with the same size as the input images $x$, aiming to optimize $\hat{x}$ following the following reconstruction objective: $l_{rec} = l_{MSE}(f(\hat{x}; w_e), s_{in})$, where $w_e$ represents the freezing weights of the extractor on the server and $s_{in}$ denotes the feature inputs.

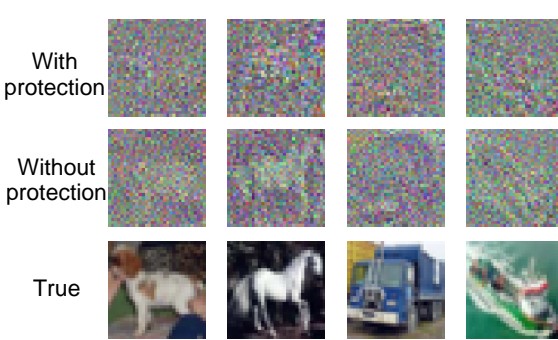

Figure 4: The comparison between privacy with protection and without protection.

To enhance user privacy within FedIN, the clients can easily **add Gaussian noise** follow the standard deviation $\sigma$ of IN feature inputs and outputs in training. Specifically, we define $\sigma_{in}$ as the standard deviation of IN feature inputs and $\sigma_{out}$ for feature outputs. The Gaussian noises are represented as $z_{in} \sim \mathcal{N}(0, \sigma_{in}^2)$ and $z_{out} \sim \mathcal{N}(0, \sigma_{out}^2)$. For simplicity in notations, we use $\sigma$ to denote $z$, as we solely adjust $\sigma$ within this privacy protection mechanism. Throughout the training phase, We apply $0.8\sigma$ to the IN feature inputs and outputs, i.e., the inputs of Eq. 5 are $\hat{s}_{i,in}^k = s_{i,in}^k + 0.8\sigma_{in}$, and for Eq. 6, they become $\hat{s}_{i,out}^k = s_{i,out}^k + 0.8\sigma_{out}$. The results of the privacy protection are shown in Figure 4, indicating the efficiency of this mechanism in protecting user privacy in FedIN. More details on the privacy experiments and discussion of Differential Privacy (DP) are further provided in Section 5.5 and Appendix D.

### 4.4 Weight Aggregation

If client models have different numbers of layers, FedIN adopts layer-wise heterogeneous aggregation (Liu et al., 2022; Chan et al., 2024), enabling the server to aggregate weights from the same layer rather than the same model. Similarly, when client models have different architectures, FedIN aggregates model weights only from models with identical architectures, the same as the homogeneous aggregation method used in FedAvg (McMahan et al., 2017) and FedDF (Lin et al., 2020). The effectiveness of FedIN with these two distinctive aggregation methods is further demonstrated in Section 5.2, and the detailed demonstrations for these two aggregation methods are shown in Appendix B.

## 5 Experiments

### 5.1 Experiment Settings

**Federated Settings.** In this section, we conduct experiments to evaluate the performances of FedIN on the CIFAR-10 (Krizhevsky et al., 2009), Fashion-MNIST (Xiao et al., 2017), SVHN (Netzer et al., 2011), and CINIC-10 (Darlow et al., 2018), which CINIC-10 is constructed from ImageNet and CIFAR-10. We establish two distributions for these datasets, independent and identically distributed (IID), and non-IID. The non-IID data is generated using a Dirichlet distribution with a parameter $\alpha = 0.5$. We have 100 clients in the FL training process. To evaluate the generalized ability of FedIN, we utilize ResNets and Vision Transformers (ViTs) separately in our experiments, which are demonstrated in Appendix E.1 with more details. We more focus on ResNets in the experiment part, and the experiments of ViTs are demonstrated in Table 2. The

Table 1: Model accuracy for IID and non-IID data of ResNets. "Round" denotes the round at which the method first achieves the target accuracy (ACC in the column head) in Non-IID. We **bold** the best results, and underline the second best results in this table.

| Methods | FashionMNIST (ACC=85) | | | SVHN (ACC=80) | | | CIFAR-10 (ACC=60) | | | CINIC-10 (ACC=50) | | |
|---|---|---|---|---|---|---|---|---|---|---|---|---|
| | IID | Non-IID | Round↓ | IID | Non-IID | Round↓ | IID | Non-IID | Round↓ | IID | Non-IID | Round↓ |
| FedAvg(2017) | 90.3 | 89.2±1.1 | 47 | 89.2 | 84.3±1.2 | 82 | 76.8 | 65.7±1.8 | 109 | 64.7 | 55.8±2.9 | 103 |
| FedProx(2020b) | 89.7 | 87.8±0.5 | 40 | 90.6 | 87.1±1.0 | 45 | 77.6 | 72.5±1.2 | 72 | 66.8 | 57.5±2.3 | 98 |
| Scaffold(2020) | 88.3 | 87.1±0.9 | 25 | 91.1 | 86.0±1.5 | 72 | 79.0 | 67.9±1.7 | 120 | 65.2 | 55.2±3.1 | 147 |
| FedNova(2020) | 87.5 | 87.3±1.4 | 36 | 87.3 | 86.5±1.7 | 106 | 62.9 | 61.4±2.5 | 229 | 62.9 | 52.3±2.7 | 207 |
| MOON(2021a) | 89.5 | 89.0±1.0 | 34 | 89.5 | 86.1±1.3 | 55 | 74.1 | 67.4±1.8 | 129 | 66.1 | 57.6±2.7 | 115 |
| HeteroFL(2021) | 89.3 | 89.5±0.6 | 140 | **93.8** | 89.3±0.6 | 107 | 72.1 | 62.5±1.3 | 273 | 63.6 | 56.1±2.1 | 183 |
| FedGen(2021) | 89.1 | 88.7±1.2 | 39 | 91.7 | 89.2±1.4 | 64 | 78.3 | 69.6±2.2 | 134 | 65.1 | 58.4±2.5 | 142 |
| FedFomo(2021) | 88.7 | 88.0±1.3 | 48 | 89.9 | 88.4±1.5 | 79 | 73.6 | 70.3±2.0 | 174 | 65.5 | 56.3±2.6 | 156 |
| FedET(2022) | 90.2 | 89.5±1.5 | 29 | 89.8 | 88.6±1.6 | 70 | 75.4 | 71.1±2.6 | 117 | 67.3 | 60.8±2.4 | 121 |
| InclusiveFL(2022) | 88.4 | 89.1±1.0 | 31 | 90.9 | 88.7±0.9 | 67 | 75.0 | 67.2±1.7 | 160 | 65.8 | 54.7±1.9 | 134 |
| FedRolex(2022) | 90.9 | 88.7±1.3 | 100 | 91.3 | 87.0±1.2 | 81 | 79.8 | 68.0±1.6 | 165 | 68.7 | 57.4±2.3 | 159 |
| ScaleFL(2023) | 91.1 | 90.1±0.7 | 95 | 93.7 | 90.2±0.7 | 100 | 76.4 | 72.0±2.0 | 108 | 69.2 | 58.1±2.4 | 120 |
| FedDPA(2023) | 90.0 | 89.4±1.2 | 30 | 90.9 | 89.6±1.4 | 58 | 77.2 | 73.5±2.2 | 96 | 68.4 | 59.4±2.3 | 93 |
| InCoAvg(2024) | 90.6 | 89.5±1.2 | 22 | 90 | 87.4±1.8 | 55 | 78.7 | 67.5±2.6 | 127 | 67.2 | 57.5±2.6 | 135 |
| FedSelect(2024) | 90.1 | 88.9±1.1 | 43 | 89.5 | 88.0±1.65 | 75 | 76.5 | 69.1±2.1 | 142 | 68.9 | 58.9±2.7 | 124 |
| FedIN | 91.2 | 90.2±1.2 | 20 | 91.8 | 89.4±1.3 | 29 | 80.5 | 74.8±2.3 | 54 | 70.1 | 61.7±2.8 | 86 |
| FedIN (+Noise) | **91.3** | **90.6±1.3** | **18** | 92.9 | **91.0±1.5** | **26** | **83.2** | **77.3±2.5** | 52 | **72.6** | **63.4±2.5** | **70** |

Table 2: Model accuracy for IID and non-IID data of ViTs. "Round" denotes the round at which the method first achieves the target accuracy (ACC in the column head) in Non-IID. We **bold** the best results, and underline the second best results in this table.

| Methods | FashionMNIST (ACC=90) | | | SVHN (ACC=90) | | | CIFAR-10 (ACC=60) | | | CINIC-10 (ACC=80) | | |
|---|---|---|---|---|---|---|---|---|---|---|---|---|
| | IID | Non-IID | Round↓ | IID | Non-IID | Round↓ | IID | Non-IID | Round↓ | IID | Non-IID | Round↓ |
| FedAvg(2017) | 93.2 | 92.4±0.9 | 20 | 93.5 | 92.3±1.0 | 13 | 94.6 | 93.7±1.5 | 14 | 84.4 | 83.2±1.9 | 20 |
| FedProx(2020b) | 93.4 | 92.2±0.7 | 18 | 93.9 | 92.8±1.1 | 14 | 95.0 | 94.0±1.3 | 13 | 84.2 | 83.3±1.6 | 23 |
| Scaffold(2020) | 92.1 | 92.5±0.8 | 23 | 92.7 | 92.0±1.3 | 17 | 94.2 | 93.6±1.5 | 21 | 83.5 | 82.9±1.5 | 26 |
| FedNova(2020) | 92.8 | 92.2±1.4 | 20 | 93.0 | 91.6±1.5 | 23 | 95.2 | 94.5±1.4 | 19 | 85.6 | 84.3±1.5 | 25 |
| MOON(2021a) | 92.5 | 91.8±1.2 | 19 | 93.8 | 93.0±1.4 | 20 | 95.4 | 94.1±1.5 | 22 | 86.2 | 84.4±1.4 | 23 |
| HeteroFL(2021) | 93.2 | 92.7±1.0 | 27 | 94.5 | 93.2±1.2 | 25 | 95.0 | 94.6±1.4 | 30 | 86.5 | 84.2±1.5 | 28 |
| InclusiveFL(2022) | 92.6 | 90.6±0.9 | 17 | 93.6 | 92.2±1.0 | 15 | 94.5 | 93.9±1.2 | 14 | 85.4 | 85.1±1.1 | 16 |
| FedRolex(2022) | 93.0 | 92.6±1.1 | 25 | 94.0 | 92.8±1.2 | 24 | 95.4 | 94.2±1.5 | 27 | 87.0 | 85.7±1.2 | 29 |
| InCoAvg(2024) | 93.2 | 92.4±1.3 | 18 | 93.8 | 92.5±1.3 | 17 | 95.1 | 94.0±1.4 | 15 | 87.3 | 85.6±1.4 | 17 |
| FedIN | **93.5** | 93.0±1.0 | 15 | 94.8 | 93.4±1.2 | 11 | 96.0 | 94.7±1.3 | 12 | **88.4** | 87.3±1.7 | 14 |
| FedIN (+Noise) | 93.4 | **93.6±1.1** | **15** | **94.9** | **94.0±1.5** | **11** | **96.0** | **95.2±1.2** | **11** | 88.1 | **87.9±1.5** | **13** |

number of communication rounds is set to 500 for ResNets and 100 for ViTs. The batch size is 16 and the sample ratio is 0.1 during the training process. For all datasets, the clients complete five epochs of local training during each communication round. Our code will be released on Github.

**Baselines.** We have 15 baselines, including two classic algorithms, FedAvg (McMahan et al., 2017) and FedProx (Li et al., 2020b), and 13 state-of-the-art methods, Scaffold (Karimireddy et al., 2020), FedNova (Wang et al., 2020), MOON (Li et al., 2021a), HeteroFL (Diao et al., 2021), FedFomo (Zhang et al., 2021), FedGen (Zhu et al., 2021), FedET (Cho et al., 2022), InclusiveFL (Liu et al., 2022), FedRolex (Alam et al., 2022), FedDPA (Yang et al., 2023), ScaleFL (Ilhan et al., 2023), FedSelect (Tamirisa et al., 2024), and InCo (Chan et al., 2024). FedIN (+Noise) is a privacy-protected version of FedIN. More discussions on user privacy are provided in Section 5.5. We use Adam optimizer with default parameter settings in PyTorch for all methods. All experiments are conducted with one Nvidia RTX3090 GPU. We show average results from three random seeds in non-IID experiments. We introduce more details of baselines in Appendix E.2.

## 5.2 Accuracy Analyses.

**Accuracy of IID and non-IID Data.** We conduct experiments on the IID and non-IID data in Fashion-MNIST, SVHN, and CIFAR-10 datasets. The experiment results are shown in Table 1 and Table 2. From Table 1, FedIN (+Noise) achieves the highest accuracy among all methods in FashionMNIST, CIFAR-10,

Table 3: Training overheads for different methods. "Params" indicates the communication overheads, and "M" means million. "Memory" refers to the memory occupied by methods in the training process.

| Metrics | Methods | | | | | | | | FedIN |
|---|---|---|---|---|---|---|---|---|---|
| | FedAvg | Scaffold | MOON | HeteroFL | FedRolex | InclusiveFL | ScaleFL | InCoAvg | |
| Params(M) ↓ | 12.28 | 24.56 | 12.28 | 16.29 | 16.29 | 12.28 | 20.65 | 12.28 | 12.35 |
| Memory(MB) ↓ | 235.0 | 470.0 | 705.0 | 445.6 | 445.6 | 235.0 | 574.8 | 235.0 | 235.3 |

Table 4: Model accuracy with heterogeneity models with FedAvg aggregations.

| Fashion-MNIST | Methods | | | | | | | | | FedIN |
|---|---|---|---|---|---|---|---|---|---|---|
| | FedAvg | FedProx | Scaffold | FedNova | MOON | InclusiveFL | FedGen | FedSelect | InCoAvg | |
| IID | 86.1 | 83.4 | 87.7 | 84.2 | 87.0 | 88.1 | 86.8 | 87.6 | 88.0 | **88.9** |
| Non-IID | 85.4 | 82.1 | 86.3 | 83.9 | 86.5 | 86.4 | 85.9 | 87.3 | 87.2 | **88.0** |

and gets the second highest accuracy in SVHN with IID. More discussions for FedIN (+Noise) are shown in Section 5.5. FedIN also gets the second-best results among different datasets. These results demonstrate the effectiveness of FedIN. Additionally, Figure 5 shows the smoothed test accuracy on non-IID data of CIFAR-10.

FedIN (red line) achieves the highest accuracy and exhibits the fastest convergent speed throughout the training process. It is the first method to achieve the target accuracy (red dot line). Moreover, FedIN incurs only a small additional overhead of one batch of feature inputs and outputs compared to FedAvg, as shown in Table 3.

**Accuracy of Homogeneous Models.** While FedIN primarily addresses the system heterogeneity challenge in FL, we also conduct experiments in a homogeneous model environment using CIFAR-10. All client models are ResNet18 in this experiment, and the remaining federated settings are the same as those in the system heterogeneity experiments. As presented in Table 5a, FedIN still outperforms state-of-the-art baselines, specifically for the baselines that are designed to enhance FL performance in homogeneous model environments.

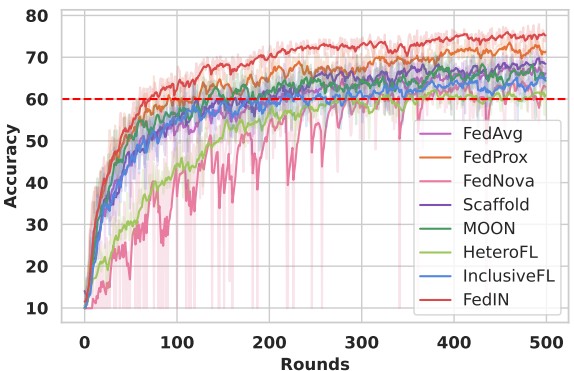

Figure 5: The smoothed test accuracy on non-IID data of CIFAR-10. The red dot line denotes the target accuracy in Table 1.

**Accuracy with FedAvg Aggregation.** It is worth noting that FedIN can be deployed in scenarios with extreme heterogeneity, where layer-wise aggregation is not feasible. In such cases, model weights can only be aggregated with models that share the same shape. Therefore, when the server aggregates weights from a specific model shape, it disregards weights from models with different shapes. This setup simulates a scenario where clients have completely different model architectures, such as CNNs and Transformers, and aggregation can only occur among clients with identical shapes. To demonstrate the effectiveness of FedIN in such extreme environments, we conducted experiments on the Fashion-MNIST dataset, utilizing FedAvg aggregation. The remaining federated settings are the same in this experiment. As indicated in Table 4, FedIN still achieves the highest accuracy, 88.9% on IID data and 88.0% on non-IID data. These results further emphasize the effectiveness of FedIN in extreme system heterogeneity environments.

## 5.3 The Reason for the Improvements

**CKA Similarity for Different Stages.** Inspired by (Luo et al., 2021) and (Raghu et al., 2021), we use CKA similarity (Kornblith et al., 2019) to examine the layer similarity among different clients across different methods. In our analysis, stage $i$ indicates the $i_{th}$ block in the ResNet architecture. We define all residual blocks with the same shape as one stage. Figure 6a and Figure 6b illustrate the CKA similarity of different

Table 5: Model accuracy with different settings in CIFAR-10.

(a) Model accuracy with homogeneous models.

| CIFAR-10 | Methods | | | | |
|---|---|---|---|---|---|
| | FedProx | Scaffold | FedNova | MOON | FedIN |
| IID | 83.5 | 84.3 | 82.0 | 84.2 | **84.7** |
| Non-IID | 77.5 | 76.8 | 75.4 | 78.2 | **79.2** |

(b) Model accuracy with ablation studies.

| CIFAR-10 | Methods | | | | |
|---|---|---|---|---|---|
| | FedAvg | w/o IN | w/o Prox | w/o Opt | FedIN |
| IID | 76.8 | 77.6 | 78.8 | 79.4 | **80.5** |
| Non-IID | 66.2 | 72.0 | 66.4 | 74.9 | **75.9** |

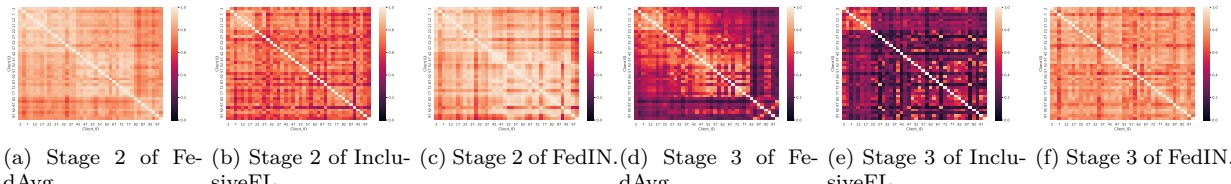

(a) CKA similarity for IID data.

(b) CKA similarity for non-IID data.

(c) Effects from different batch sizes.

(d) Effects from different sample numbers.

Figure 6: Illustrations for CKA similarity of IID data in Figure 6a and non-IID data in Figure 6b with CIFAR-10. The effects from different batch sizes and different sample numbers are shown in Figure 6c and Figure 6d under non-IID CIFAR-10.

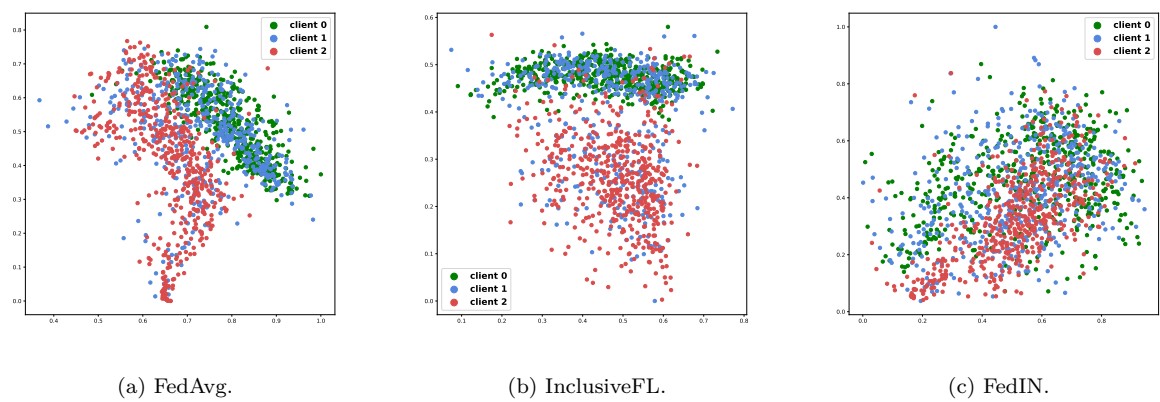

(a) Stage 2 of FedAvg.

(b) Stage 2 of InclusiveFL.

(c) Stage 2 of FedIN.

(d) Stage 3 of FedAvg.

(e) Stage 3 of InclusiveFL.

(f) Stage 3 of FedIN.

Figure 7: Heatmaps of CKA similarity from stage 2 and 3 among different clients in CIFAR-10.

(a) FedAvg.

(b) InclusiveFL.

(c) FedIN.

Figure 8: T-SNE visualization of features learned by different methods from stage 3 on CIFAR-10. We select data from the same class and utilize three models with different architectures (Client0: ResNet10, Client1: ResNet14, Client2: ResNet26).

stages under IID and non-IID. Notably, in these results, FedIN exhibits the highest similarity even in the deepest stage (stage 3), while FedAvg and InclusiveFL struggle to maintain high similarity levels in stage 3, as evidenced by the gray area in the figure. To gain further insights into the dissimilarities between FedIN and the other methods, we present heatmaps of similarity from stage 2 and stage 3 among clients in Figure 7, indicating that the average similarity of FedIN surpasses that of FedAvg and InclusiveFL. These results and analyses suggest that FedIN ensures consistency among the deep layers of client models, indicating that client models learn more general knowledge in FedIN compared to other baselines.

Table 6: Model accuracy with different client numbers on CIFAR-10.

| Methods | IID | | | | | Non-IID | | | | |
|---|---|---|---|---|---|---|---|---|---|---|
| | $N_c = 10$ | $N_c = 20$ | $N_c = 50$ | $N_c = 100$ | $N_c = 200$ | $N_c = 10$ | $N_c = 20$ | $N_c = 50$ | $N_c = 100$ | $N_c = 200$ |
| FedAvg | 79.3 | 79.2 | 78.7 | 76.8 | 74.0 | 68.3 | 67.9 | 66.9 | 66.2 | 62.5 |
| InclusiveFL | 77.5 | 76.7 | 79.1 | 75.0 | 73.4 | 66.8 | 68.4 | 67.1 | 66.1 | 61.2 |
| FedIN | **82.8** | **83.1** | **81.0** | **80.5** | **74.3** | **76.7** | **76.3** | **74.1** | **75.9** | **72.2** |

**T-SNE Visualization.** We conduct t-SNE visualizations (Van der Maaten & Hinton, 2008) on features extracted from stage 3 in Figure 8, focusing on data belonging to the same class. The objective is to observe the clustering behavior of these data points. In Figure 8a and Figure 8b, it is evident that the features from client 0 and client 1 and features from client 2 are separated. However, the features from these three clients form a singular cluster in FedIN, as depicted in Figure 8c, validating that the features from data with the same class from different model architectures are consistent.

## 5.4 Ablation Study

We conduct an ablation study to evaluate the contributions of the key components in FedIN. Our ablation study includes the following methods: (i) FedAvg, (ii) FedIN w/o IN (FedIN without IN loss), (iii) FedIN w/o Prox (FedIN without Prox regularized term), (iv) FedIN w/o Opt (FedIN without the gradient alleviation (optimization)). Table 5b and Figure 9 illustrates the results of the ablation studies.

**Effects of the Gradient Alleviation and the Loss Function.** In this experiment, we highlight that our solution is advantageous and effective in solving the gradient divergence problem. Figure 9 illustrates the results of considering the gradient divergence problem and ignoring this problem.

The accuracy achieved by FedIN surpasses that of FedIN w/o Opt, and the convergent speed of FedIN is also accelerated, as observed in Figure 9. Moreover, after 200 rounds, FedIN w/o Prox becomes unstable and its performance deteriorates during the subsequent training process. At last, FedIN w/o Prox only achieves the performance like FedAvg, as shown in Table 5b, hinting that the improvement from IN loss is eliminated at the end of the training process. Therefore, the inclusion of a regularized term becomes essential to maintain the effectiveness of IN loss throughout the training process.

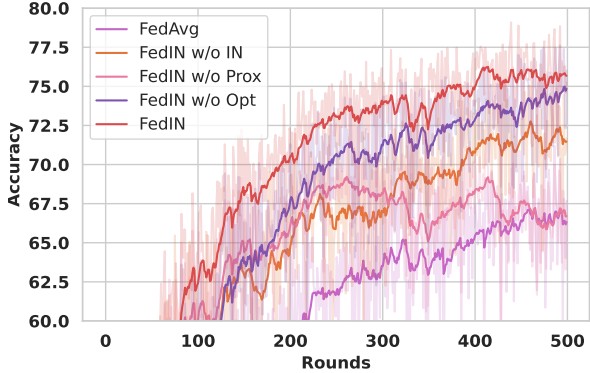

Figure 9: Smoothed test accuracy for non-IID data of CIFAR-10 in the ablation study.

**Effects of Client Numbers, Batch Sizes, and Sample Numbers.** To investigate the effects of varying client numbers, we conduct experiments on CIFAR-10, as presented in Table 6. $N_c$ denotes the number of clients. Notably, FedIN outperforms the other methods across different numbers of clients. We also conduct analysis on different batch sizes and sample numbers on CIFAR-10 to verify the effects of these hyperparameters. As shown in Figure 6c, batch sizes 16, 32, and 64 are the best selections, but the batch sizes of 8 and 128 still outperform HeteroFL and InclusiveFL. Considering the communication overhead, a batch size of 16 is the optimal choice. From Figure 6d, it is clear that increasing the sample numbers has little impact on accuracy improvement.

## 5.5 Privacy Analysis

We assume that the server reconstructs user images from IN inputs, originating from the outputs of the first layer, as images are more easily reconstructed from features extracted from shallow layers. The experimental results presented in Table 7 and Figure 10 provide a comprehensive analysis of the impact of how privacy-preserving mechanisms, adding Gaussian noise, ensure user privacy. We use LPIPS (Balle et al., 2022) to represent the quality of reconstruction images.

Table 7: Model accuracy with adding noise in FedIN to protect privacy.

| Dataset | Noise Level (+FedIN) | | | | | | | | |
|---|---|---|---|---|---|---|---|---|---|
| | w/o Noise | $+0.1\sigma$ | $+0.2\sigma$ | $+0.5\sigma$ | $+0.8\sigma$ | $+1.0\sigma$ | $+2.0\sigma$ | $+3.0\sigma$ | $+5.0\sigma$ |
| SVHN | 89.3 | 90.1 | 90.2 | 90.0 | **91.0** | 89.2 | 83.7 | 79.8 | 65.3 |
| Fashion-MNIST | 90.3 | 89.8 | 90.4 | 90.6 | **90.6** | 90.4 | 88.7 | 84.9 | 69.4 |
| CIFAR-10 | 75.9 | 75.2 | 76.4 | 77.2 | **77.3** | 76.2 | 73.5 | 70.0 | 30.6 |

(a) Similarities of reconstruction images.

(b) Reconstruction images and ground truths.

Figure 10: The results from the reconstruction experiments. Figure 10a evaluates the quality of reconstruction images, using LPIPS (Balle et al., 2022), which lower value indicates more clear images. Figure 10b illustrates the reconstructed images from protected IN inputs and original IN inputs.

**Accuracy with Different Noise Levels.** Table 7 illustrates the impact of adding noise at varying levels in FedIN on the model accuracy. Moderate noise levels, especially at $+0.8\sigma$ (the setting of FedIN (+Noise)), obtain superior performance compared to FedIN without adding noise. However, as noise levels exceed $+1.0\sigma$, there is a noticeable decline in accuracy across all datasets, indicating a degradation in model performances due to excessive noises. These results suggest that appropriate noise levels can protect user privacy while aiding in model generalization. The features with adding appropriate noises enrich the feature spaces, implying more diversified $s_{in}$ and $s_{out}$ for IN training.

**Reconstruction Images.** In Figure 10, we add $0.8\sigma$ to IN inputs for privacy protection. The LPIPS of reconstruction images from models trained with privacy protection stabilizes at a higher value (0.18) compared to those trained without protection, as shown in Figure 10a. Figure 10b displays the reconstructed images at the server using these two methods. These results indicate that the server encounters challenges in reconstructing user images when IN inputs are protected.

## 6 Limitations and Conclusions

**Limitations:** Limitations are that FedIN cannot directly manage the models with different widths, leading to different sizes of $s_{in}$ and $s_{out}$, and also the models from different families, such as CNNs and Transformers. We have two ways to alleviate these environments in FedIN. One method is aggregating model weights only from models with identical shapes, for which we have conducted experiments in Section 5.2. The other method involves utilizing linear layers to project the dimension of $s_{in}$ and $s_{out}$ to the same size, enabling the deployment of IN training after these projections. Specifically, we can deploy layers in the models similar to early exits, mentioned in ScaleFL Ilhan et al. (2023), to be the projected layers in FedIN. We will investigate this method further in our future work.

**Conclusions:** We propose a method called FedIN, which conducts local training based on the private dataset and IN training from the client features, requiring only one batch of features. Moreover, we formulate a convex optimization problem to tackle the gradient divergence problem. To protect user privacy, we further propose a simple yet effective method, adding Gaussian noise during the IN training process. We conduct extensive experiments on four public datasets with 15 baselines to demonstrate the superior performances of FedIN.

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

## A    Derivation for Convex Optimization Problem

In Section 4.2, we propose the convex optimization problem to alleviate the gradient divergence caused by local training and IN training, which is shown as follows,

$$\min_Z ||G_{IN} - Z||_F^2, \quad s.t. \ \langle Z, G_{local} \rangle \geq 0, \tag{14}$$

which we solve this convex problem by the Lagrange dual problem (Bot et al., 2009; Chan et al., 2024). First, the Lagrangian of Eq. 14 is illustrated as,

$$
\begin{aligned}
L(Z, \lambda) =& tr((G_{IN} - Z)^T(G_{IN} - Z)) - \lambda tr(Z^T G_{local}) \\
=& tr(G_{IN}^T G_{IN} - Z^T G_{IN} - G_{IN}^T Z + Z^T Z) - \lambda tr(Z^T G_{local}) \\
=& tr(G_{IN}^T G_{IN}) - 2tr(Z^T G_{IN}) + tr(Z^T Z) - \lambda tr(G_{local}^T Z).
\end{aligned}
\tag{15}
$$

To get the optimum point for $L(Z, \lambda)$, we set $\frac{\partial L(Z, \lambda)}{\partial Z} = 0$, and then we get

$$Z = G_{IN} + \lambda G_{local}/2, \tag{16}$$

which is the optimum point for $L(Z, \lambda)$ and also for the primal problem Eq. 14. Moreover, the dual problem is obtained by $L(\lambda) = \inf_Z L(Z, \lambda)$. We replace $Z = G_{IN} + \lambda G_{local}/2$ in $L(Z, \lambda)$ to get $L(\lambda)$ as shown below,

$$
\begin{aligned}
L(\lambda) =& tr(G_{IN}^T G_{IN}) - 2tr((G_{IN} + \frac{\lambda G_{local}}{2})^T G_{IN}) + tr((G_{IN} + \frac{\lambda G_{local}}{2})^T (G_{IN} + \frac{\lambda G_{local}}{2})) \\
& - \lambda tr((G_{IN} + \frac{\lambda G_{local}}{2})^T G_{local}) \\
=& tr(G_{IN}^T G_{IN}) - 2tr(G_{IN}^T G_{IN}) - \lambda tr(G_{local}^T G_{IN}) + tr(G_{IN}^T G_{IN} + \frac{\lambda G_{local}^T G_{IN}}{2} + \frac{\lambda G_{IN}^T G_{local}}{2} \\
& + \frac{\lambda^2 G_{local}^T G_{local}}{4}) - \lambda tr(G_{IN}^T G_{local}) - \frac{\lambda^2 tr(G_{local}^T G_{local})}{2} \\
=& tr(G_{IN}^T G_{IN}) - 2tr(G_{IN}^T G_{IN}) + tr(G_{IN}^T G_{IN}) - \lambda tr(G_{local}^T G_{IN}) + \lambda tr(G_{local}^T G_{IN}) \\
& + \frac{\lambda^2 tr(G_{local}^T G_{local})}{4} - \frac{\lambda^2 tr(G_{local}^T G_{local})}{2} - \lambda tr(G_{IN}^T G_{local}) \\
=& -\frac{\lambda^2}{4} tr(G_{local}^T G_{local}) - \lambda tr(G_{local}^T G_{IN}).
\end{aligned}
\tag{17}
$$

Therefore, we have the dual problem, as follows,

$$\max_\lambda \ L(\lambda) = -\frac{\lambda^2}{4} tr(G_{local}^T G_{local}) - \lambda tr(G_{local}^T G_{IN}), \quad s.t. \ \lambda \geq 0, \tag{18}$$

then we complete the derivation process for convex optimization problem using Lagrange dual problem.

## B    Details for Weight Aggregation

In Section 4.4, we present two aggregation methods for FedIN. One is layer-wise heterogeneous aggregation (Liu et al., 2022; Chan et al., 2024), and the other one is FedAvg aggregation from FedAvg (McMahan et al., 2017) and FedDF (Lin et al., 2020). In this section, we demonstrate the details of these two aggregation methods.

### B.1    Layer-wise Heterogeneous Aggregation

To facilitate heterogeneous models in FL, we utilize layer-wise aggregation. Specifically, the weights of the same layer from different models would be averaged. For example, Figure 11a illustrates the heterogeneous aggregation for $n$ groups, $s_1, ..., s_n$. The clients in the same group share the same model architecture. $w_{l_i, s_j}$

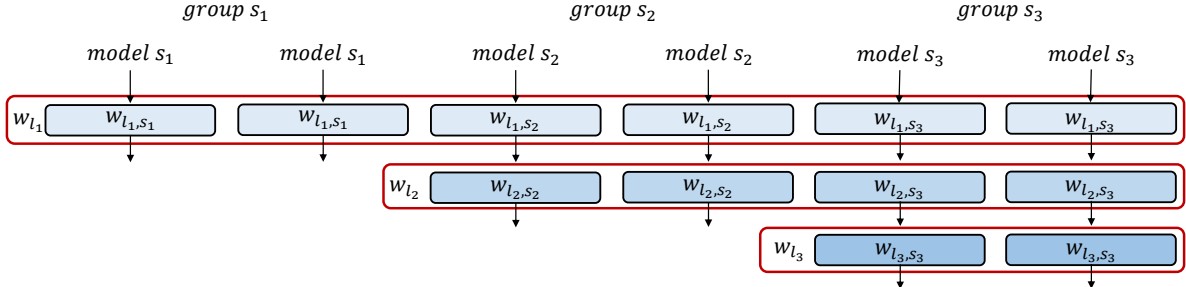

(a) Layer-wise heterogeneous aggregation.

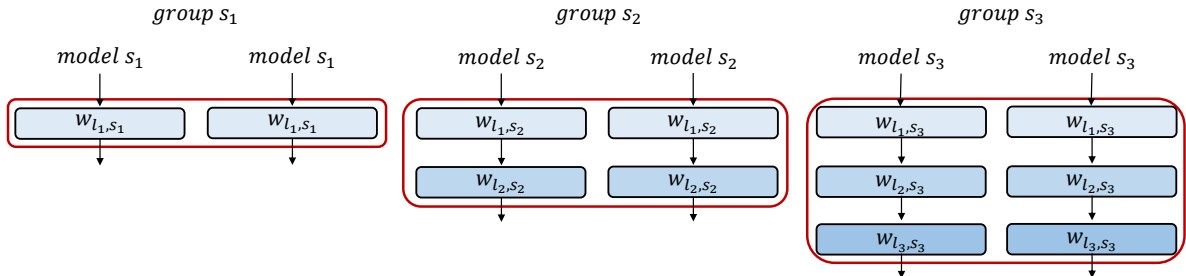

(b) FedAvg aggregation with heterogeneous models.

Figure 11: Two aggregation methods for FedIN: (a) Figure 11a: Layer-wise heterogeneous aggregation. This method aggregates model weights layer by layer. (b) Figure 11b: FedAvg aggregation with heterogeneous models. This method aggregates model weights only from models with identical shapes.

denotes the weights of layer $i$ in group $s_j$. We use a superscript to denote that $w^k_{l_i,s_j}$ belongs to the $k$-th client in group $s_j$. The group indexes are ordered by the size of the model architectures, i.e., the size of the client models in group $s_1$ is the smallest. As shown in the example in Figure 11a, all groups share layer 1, denoted by $w_{l_1,s_j}, j = 1, ..., n$. Then, layer $w_{l_i}$ only appears in group $s_j$ when $i \leq j$. Following the above definition, the heterogeneous aggregation for layer $w_{l_i}$ is expressed as follows,

$$w_{l_i} = \frac{\sum_{j=i}^{n} \sum_{k=1}^{|s_j|} w^k_{l_i,s_j}}{\sum_{j=i}^{n} |s_j|}, \tag{19}$$

where $|s_j|$ denotes the number of clients in group $s_j$.

## B.2   FedAvg Aggregation

When client models have different shapes, FedIN aggregates model weights exclusively from models with identical shapes, like the aggregation method employed in FedAvg, as illustrated in Figure 11b. Similarly, the environment has $n$ different model groups, denoted as $s_1, ..., s_n$. For layer $w_{l_i,s_j}$ in group $s_j$, this aggregation method is defined as follows,

$$w_{l_i,s_j} = \frac{\sum_{k=1}^{|s_j|} w^k_{l_i,s_j}}{|s_j|}. \tag{20}$$

Considering different sizes for local datasets in group $s_j$ and simplifying the notation $l_i$ for model weights in this method, the aggregation method for $w_{s_j}$ in group $s_j, j = 1, ...n$ is formulated as follows,

$$w_{s_j} = \frac{\sum_{k=1}^{|s_j|} d_k w^k_{s_j}}{\sum_{k=1}^{|s_j|} d_k}, \tag{21}$$

where $d_k$ is the size of local datasets for client $k$ in group $s_j$.

## C  Compared with sub-model methods

FedIN distills knowledge from different sub-models between the extractors and the classifiers in each client. We focus on the feature alignments for sub-models, according to distillation for different feature pairs. However, FedIN differs fundamentally from sub-model methods, like HeteroFL Diao et al. (2021) and FedRolex Alam et al. (2022). While sub-model methods primarily address different model-splitting strategies and aggregation approaches during training, FedIN employs a distinct methodology by utilizing distillation for feature pair alignment.

## D  Differential Privacy for Privacy Consideration

In this section, we discuss how our privacy consideration method, introduced in Section 4.3, follows differential privacy (DP). First, we present the following assumption, which is frequently used in differential privacy of federated learning (FL) Wei et al. (2020).

**Assumption D.1.** (Upper bound of client features). *Given a function $f_i : \mathcal{D}_i \to \mathbb{R}^d$ defined on client dataset $\mathcal{D}_i$ of client $i$, the upper bound of client features satisfies that $\|f_i(x_i)\|_2 \leq C$, where $x_i \in \mathcal{D}_i$ and $C$ is a constant.*

Based on Assumption D.1, the $\ell_2$-sensitivity of $f$ is $\Delta_2(f) = \max_{D_i, D_j} \|f(D_i) - f(D_j)\|_2 \leq 2C$, where $D$ and $D'$ are neighboring datasets. According to Dwork et al. (2014), a Gaussian mechanism $\mathcal{N}(0, \sigma^2)$ that follows,

$$\sigma \geq \frac{\Delta_2 f \sqrt{2\ln(1.25/\delta)}}{\epsilon}, \tag{22}$$

can be used to guarantee the $(\epsilon, \delta)$-DP. In our method, we consider adding noise distributions $z \sim \mathcal{N}(0, 0.8\sigma^2)$ to feature inputs and outputs. To ensure $(\epsilon, \delta)$-DP in FedIN, we can select appropriate values for $\epsilon, \delta, C$. First, from Assumption D.1, we can obtain $\Delta_2(f) \leq 2C$, then

$$\sigma \geq \frac{2C\sqrt{2\ln(1.25/\delta)}}{\epsilon} \geq \frac{\Delta_2 f \sqrt{2\ln(1.25/\delta)}}{\epsilon}. \tag{23}$$

Therefore, by controlling the values of $\epsilon, \delta, C$, we can determine $\sigma$ in our Gaussian mechanism to ensure $(\epsilon, \delta)$-DP. For example, we set $C = 0.1$, $\epsilon = 10$ and $\delta = 10^{-6}$, then we get $\sigma \geq 0.127$.

## E  More Experiments

### E.1  Details of Model Architectures

The model architectures of ResNets are ResNet10, ResNet14, ResNet18, ResNet22, and ResNet26 from PyTorch source codes, and of ViTs are ViT-S/8, ViT-S/9, ViT-S/10, ViT-S/11, and ViT-S/12. Five different model architectures are evenly distributed among 100 clients. We use one CNN layer at the beginning as an extractor, and one MLP layer at the end for a classifier in ResNets. The first ViT block in ViTs is an extractor, and the last MLP layer at the end is a classifier in ViTs. More details of the extractor, intermediate layers, and the classifier are shown in Figure 12.

### E.2  Details of Baselines

HeteroFL, InclusiveFL, FedRolex, ScaleFL, and InCoAvg are the methods that focus on heterogeneous models in FL. FedFomo, FedDPA, and FedSelect are the personalized FL methods. Except for the methods concentrated on the heterogeneous models, all other methods utilize the layer-wise aggregation technique proposed in (Chan et al., 2024; Liu et al., 2022) under our heterogeneous model environment. Since HeteroFL, FedRolex, and ScaleFL require model splitting based on their own methodologies, they cannot utilize this aggregation technique. To maintain a similar number of parameters as the other baselines, we deploy ResNet152 in these baselines instead of using the largest model, ResNet26, as in other methods. The model split mode in these baselines is "dynamic_a1-b1-c1-d1-e1" from the source code because of five heterogeneous models in all other methods. The hyper-parameter $\frac{\mu}{2}$ for FedProx and FedIN is 0.05.

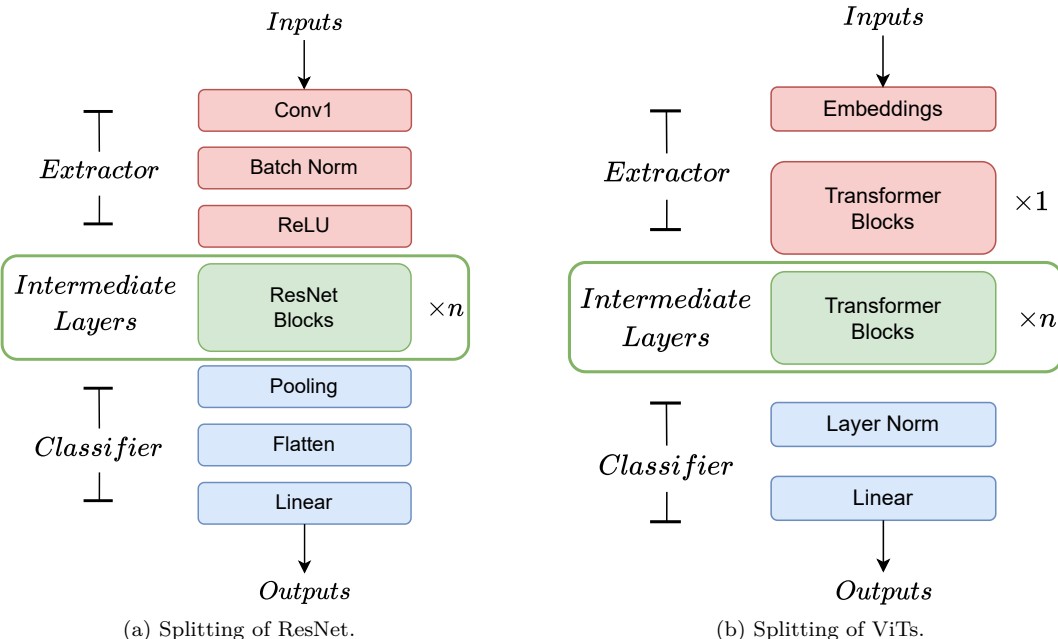

(a) Splitting of ResNet.       (b) Splitting of ViTs.

Figure 12: Different splitting ways for different model architectures: (a) Figure 12a: Client models have different numbers of the ResNet Blocks (intermediate layers), which are ResNet10 to ResNet26 in our experiments. (b) Figure 12b: Client models have different numbers of the Transformer Blocks (intermediate layers), which are ViT-S/8 to ViT-S/12 in our experiments.

