# OpenReview forum: "FedIN: Federated Intermediate Layers Learning for Model Heterogeneity"
_TMLR — Rejected by TMLR_

### Review · Reviewer_kdcV · 2025-04-22

**Summary Of Contributions:**

This submission introduces FedIN, a new method to address system heterogeneity in Federated Learning (FL) through heterogeneous model training. The motivation stems from the practical challenge that enforcing a uniform global architecture across clients is often infeasible due to varying computational capabilities.
FedIN leverages Intermediate Layer (IN) training, which aligns the output features of the central portion of the model - i.e., between the input and final classification layers - which can indeed be heterogeneous among clients, i.e. composed by a different number of layers. The number of layers in each client's local model can be tailored to reflect their device’s compute capacity, enabling flexible participation in FL.
The method is evaluated on standard FL benchmarks (CIFAR-10, FMNIST, SVHN) in image classification. Results demonstrate that FedIN is effective in handling architectural heterogeneity. Despite the additional communication cost from transmitting intermediate representations, the method compensates with faster convergence and achieves higher final accuracy.

**Audience:**

Yes

**Claims And Evidence:**

No

**Requested Changes:**

# Critical Changes
**1. Clarify experimental setup and client (system) heterogeneity.** Clearly describe how model architectures are assigned to clients and provide rationale for these choices. Indicate whether the method assumes full participation or supports partial participation. If possible, include empirical results under partial participation to assess robustness in more realistic FL settings.\
**2. Justify key design decisions.** Provide motivation for performing IN training on the client side rather than on the server, especially considering potential benefits such as reduced communication, increased privacy, and lower client computation. The use of the proximal penalty also requires justification, particularly since FedProx does not outperform FedAvg in the experiments. Additionally, consider exploring alternative losses for IN training (e.g., KL divergence instead of MSE).\
**3. Extend experiments to more tasks and realistic datasets.** The current evaluation is limited to small-scale image classification tasks. To better demonstrate the practical relevance and scalability of FedIN, please include experiments on larger benchmarks (e.g., Landmarks-User120K [1]) and other domains such as NLP. For example, the use of DistilBERT on federated text classification tasks (as in [2]) could help assess generalizability outside image classification.\
**4. Detail hyperparameter tuning process.** Include comprehensive information about how hyperparameters (learning rate, local epochs/steps, regularization coefficients, etc.) were chosen for both FedIN and baselines. Add a sensitivity analysis to show how performance varies with these choices, particularly given that reported accuracy improvements are relatively small ($\approx 1\\%$).\
**5. Ensure reproducibility and statistical rigor.** Many results are presented without measures of variability, making it difficult to assess the statistical significance of reported gains. Please include standard deviations or confidence intervals across all tables and figures to improve interpretability and reliability.\
**6. Clarify comparison of communication/computational cost w.r.t other algorithms.** It is not clear why Tables 2-3 list comparisons with only some of the competitors algorithms considered in the study. Please consider all the algorithms that tackle heterogeneous model training  (most importantly InCo, InclusiveFL and ScaleFL), highlighting how calculations has been made.\
**7. Strengthen conceptual framing connecting with distillation and submodel extraction.** The effectiveness of IN training likely builds on principles from model distillation—specifically, matching intermediate representations between models of varying depths. Highlighting this connection would help clarify why the method works and how it builds on established ideas, especially since FedIN performs feature matching across multiple heterogeneous models rather than a fixed teacher-student pair. Additionally, a clear discussion with methods that perform a different but related form of submodel extraction is missing and should be provided. I also believe that FjORD should be explicitly considered, especially given the relationships highlighted in W5.\


# Proposed Changes
**1. Compare FedIN with a data-shared baseline.** It would be interesting as an analysis to compare the performance of using the feature matching logic of FedIN with explicit use of actual clients data to perform the model heterogeneous training (which we should expect to perform better). The gap between the two should give the idea of the loss we have from not having access data to shared data. This is important because the work is motivated by the difficulty of collecting public data similar to the training data distribution.\
**2. Include an ablation study on the IN loss function.** The current implementation uses MSE for the feature matching loss. It would be valuable to explore whether alternative objectives - such as KL divergence.\
**3. Discuss integration with classical FL optimization algorithms.** Analyze the compatibility of FedIN with existing methods designed to mitigate statistical heterogeneity, such as SCAFFOLD or FedDyn. A brief discussion or experimental validation would clarify whether FedIN complements or conflicts with these strategies.\
**4. Add a pseudo-code of FedIN.** Including a clear and concise pseudo-code for the FedIN algorithm—preferably in the main text—would significantly improve the clarity and reproducibility of the method.\

[1] Hsu et al., Federated Visual Classification with Real-World Data Distribution, ECCV 2021\
[2] Nguyen et al., Where to Begin? On the Impact of Pre-Training and Initialization in Federated Learning, ICLR 2022\
[4] Romero et al., FitNets: Hints for Thin Deep Nets, ICLR 2015\
[5] Zhang et al., Be Your Own Teacher: Improve the Performance of Convolutional Neural Networks via Self Distillation, ICCV 2019\
[6] Tung et al., Similarity-Preserving Knowledge Distillation, ICCV 2019\
[7] Beyer et al., Knowledge distillation: A good teacher is patient and consistent, CVPR 2022\
[8] Sanh et al., DistilBERT, a distilled version of BERT: smaller, faster, cheaper and lighter, NeurIPS 2019\
[9] Wu et al., FIARSE: Model-Heterogeneous Federated Learning via Importance-Aware SubModel Extraction, NeurIPS 2024\
[10] Ba et al., Do Deep Nets Really Need to be Deep?, NeurIPS 2014

**Strengths And Weaknesses:**

# Strengths
**1. Well-Motivated Methodology Inspired by Distillation.** The idea underlying the idea of the method is clear and drawn from the literature of model knowledge distillation - matching features of subparts of a model architecture composed of different numbers of layers. The idea of using output feature matching is conceptually similar to a scenario where a public dataset (of input-output features pairs) is available at the server, I believe this is the main reason underlying the method (disregarding for a moment the fact that feature matching is effectively done with different model subparts).\
**2. Clear Method Description and Broad Baseline Coverage.** The explanation of how the method works is mostly clear and images provide some intuitive visualization of the transmission and local training process. Additionally, the comparison with the state of art covers numerous methods, although their explanation and organization should be improved.

# Weaknesses
**1. The assumptions underlying the method are not clearly articulated, which impacts the overall clarity of the paper.** From my understanding, the key assumption for the method’s success is that the central part of the network, even with varying numbers of layers, can still produce similar features [10] (i.e., low MSE, the loss used for IN training). This is conceptually similar to model distillation, where knowledge is transferred from a larger model to a smaller one (e.g., [4,5,6,7,8]), but here the focus is solely on the intermediate layers. A key difference is that FedIN involves heterogeneous model aggregation, meaning it operates in parallel across multiple model sizes, which introduces a more complex aggregation process. I believe both aspects - the similarity with model distillation/compression and the use of multiple depths in parallel - warrant a more detailed discussion and validation. The former connects to established concepts in the literature, while the latter represents the core technical novelty of the method. To better understand the validity of this approach, I suggest conducting further analyses, such as examining the average deviation of output features as the number of layers in the central part of the network varies, and investigating how this changes with model size allocation across clients (which would reflect the "skewness" of client computational capabilities).\
**2. Some design choices in the algorithm are not sufficiently motivated.** The addition of the proximal penalty during local training is not fully justified, especially given that, in the experiments, FedProx performs worse than FedAvg, even in the non-iid setting used to highlight the impact of the proximal penalty (Fig. 9). Moreover, it is not clear why the IN training is conducted client-side rather than server-side, which could potentially lead to significant communication savings.
I believe it would be valuable to consider performing IN training on the server side, and in that case the proximal penalty would help balance the main objective (i.e. the classification loss, first term in the right-hand side of Eq. (3)) and model distillation/compression objective (i.e. the feature matching loss (Eq. (7)). This would also allow for a simpler algorithm, reduce computational load on clients (a key concern in the context of this work), lower communication costs, and enhance privacy (since clients would not receive features from other clients). Additionally, the sensitivity to hyperparameters such as batch size for IN training could be reduced. It would also be beneficial to consider alternatives to the MSE loss for IN training, such as the KL divergence: currently the paper does not provide convincing motivation to completely disregard this possibility.\
**3. A conceptual comparison with approaches using submodel extraction is missing.** The similarities and differences between FedIN and such approaches should be clearly outlined, both in section 2.2 and when discussing the experimental results in section 5. In particular, the IN training appears to be an alternative method for reducing the number of parameters by cutting the model layerwise rather than adjusting the width of layers. In this sense, the proposed algorithm performs submodel extraction similarly to FjORD, by sampling a portion of the layers instead of a portion of parameters of each layer.
Authors should discuss pros and cons of both kinds of approaches and explain how they conceptually differ. In this light, I believe it is needed to compare “FedIN w/o IN” with FjORD, to appreciate the difference between selecting a subset of layers or subset of parameters for each layer. In the discussion of related works, please also consider more recent methods, such as [9].\
**4. The paper lacks sufficient reproducibility details**, raising concerns about the fairness of the comparisons made, particularly since the accuracy gains are often modest (around $1\\%$). There is insufficient information regarding the experimental setup, such as how models of different sizes are assigned to clients, how the metrics in Table 2 are computed, and whether the networks are pretrained or trained from scratch. Moreover, the paper does not clarify how the method performs when fine-tuning pretrained networks via FL training—a common scenario when pretrained models are available (e.g., [2]).\
**5. The experimental scope is limited, both in terms of the tasks and complexity of the datasets used.** Given that the paper’s contribution is primarily experimental, and considering the relatively small performance gains in simpler scenarios (as well as the lack of detailed tuning and training process explanations), this limitation affects the robustness of the findings. Authors should consider employing more complex larger-scale datasets to present more convincing evidence of the method’s effectiveness (see [1]).\
**6. The results presented in Fig. 10b are not fully convincing.** The poor reconstruction results without any protective measures raise concerns about the reconstruction algorithm's ability to produce meaningful outputs, more so than the success of the protection mechanism itself. Further clarification of this issue would help strengthen the results. Moreover, it is not clear if figures 4 and 10b depict the same kind of experiment, this should be clarified and redundant images eventually removed.

---

> ### Author Response · Authors · 2025-05-22
> **Response to Reviewer kdcV [1/2]**
>
> Thank you for your valuable and constructive review. We now address your main concerns as follows, and the updated parts are highlighted in blue.
>
> **Critical Changes 1.** Clarify experimental setup and client (system) heterogeneity.
>
> **Answer:** We sincerely apologize for the lack of experimental details. Our experiments are conducted under partial participation with a sampling ratio of 0.1, meaning the server randomly selects 10 clients per communication round. We have added this clarification in Section 5.1. Additionally, we used both ResNets and Vision Transformers (ViTs) as client models since these architectures represent fundamental and widely adopted models for vision tasks. The results obtained with these models are therefore highly representative.
>
> **Critical Changes 2.** Justify key design decisions. Provide motivation for performing IN training on the client side rather than on the server, especially considering potential benefits such as reduced communication, increased privacy, and lower client computation. The use of the proximal penalty also requires justification, particularly since FedProx does not outperform FedAvg in the experiments. Additionally, consider exploring alternative losses for IN training (e.g., KL divergence instead of MSE).
>
> **Answer:**
> - **IN training on the server:** A critical challenge is that the server can not access to client user data. If we perform IN training on the server, the model would only train on IN features without actual local data. These IN features serve as anchors during client training with local data in FedIN. Without these anchors, clients cannot properly align intermediate layers when training solely on local datasets.
> - **Proximal penalty:** As demonstrated in Fig. 9, the proximal penalty helps stabilize FedIN's training process and improves performance. Without this term in the loss function, the training converges to suboptimal solutions, resulting in lower accuracy (see "FedIN w/o Prox" results in Fig. 9).
> - **Using MSE as loss function: ** Since $s_{in}$ and $s_{out}$ are not probability distributions, incorporating other loss functions like KL divergence becomes challenging, as KL divergence requires probability distributions. We validate this statement through simple experiments on Non-IID datasets, as shown in the following table.
>
> | Loss function | CIFAR10 | FashionMNIST | SVHN | CINIC-10 |
> | -------- | -------- | -------- | -- | - |
> | MSE     | 74.8     | 90.2    | 89.4 | 61.7 |
> | KL-divergence | 70.1 | 88.5 | 88.0 | 56.7 |
>
>
> **Critical Changes 3.** Extend experiments to more tasks and realistic datasets.
>
> **Answer:** Thank you for your comments. In our experiments, CINIC-10 is also a large-scale dataset, containing 90,000 training images. However, Due to the limitations of our machines, we are unable to extend our experiments with additional clients and larger datasets. We plan to conduct these more extensive experiments for FedIN using cloud resources in future work.
>
> **Critical Changes 4. & 5.** Detail hyperparameter tuning process. & Ensure reproducibility and statistical rigor.
>
> **Answer:** Thank you for your valuable comments. We use the default Adam optimizer settings for our client optimizers. The experimental settings consist of 100 clients, 5 local epochs, and a sampling ratio of 0.1. The sensitivity analysis results are presented in Fig.6 (c) and (d) and Table 5. Additionally, we will release our source code to ensure the reproducibility.
>
> **Critical Changes 6.** Clarify comparison of communication/computational cost w.r.t other algorithms.
>
> **Answer:** Thank you for your valuable comments. We have updated the results for other methods for communication and computational resources. We consider the averaged transmitted parameters for all different model architectures in each communication round. Moreover, we have conducted additional experiments for Table 3. Furthermore, regarding Table 3, not all methods can employ the FedAvg aggregation method. For instance, HeteroFL and FedRolex utilize different model architectures across rounds, making FedAvg aggregation incompatible.

---

> > ### Author Response · Authors · 2025-05-22
> > **Response to Reviewer kdcV [2/2]**
> >
> > **Critical Changes 7.** Strengthen conceptual framing connecting with distillation and submodel extraction.
> >
> > **Answer:**
> > - Thank you for bringing up this suggestion. In FedIN, we distill knowledge from different sub-models between the extractors and the classifiers in each client. We focus on the feature alignments for sub-models, according to distillation for different feature pairs.
> > - However, FedIN differs fundamentally from sub-model methods. While sub-model methods primarily address different model-splitting strategies and aggregation approaches during training, FedIN employs a distinct methodology, using distillation for feature pairs. We have clarified this distinction in the revised manuscript in Appendix C.
> > - Additionally, we have evaluated three sub-model extraction methods in our experiments (HeteroFL, FedRolex, and ScaleFL), which represent established approaches similar to FjORD.

---

### Review · Reviewer_r8oD · 2025-04-23

**Summary Of Contributions:**

This paper proposes FedIN, a new method consists of local training and IN training across devices for improving model performance in heterogeneous FL data and systems. Authors conducted amounts of experiments for algorithm comparison, providing analysis and discussion for privacy protection and gradient divergence.

**Audience:**

Yes

**Claims And Evidence:**

Yes

**Requested Changes:**

See Weaknesses.

**Strengths And Weaknesses:**

**Strengths**
1. The paper is well-written and easy to read.
2. This paper conducted extensive experiments, including baseline comparison, multiple dataset experiments in both ResNet and ViTs, ablation studies.
3. This paper discussed privacy protection problem, which is a big concern in FL, especially in terms of sharing features across clients

**Weaknesses**
1. The motivation of this paper is somewhat unclear to me. In IN training, the intermediate layers are designed to convert local data features into those sampled from other clients. Is the goal to enhance the feature representation capability of the intermediate layers? If so, why not train a separate aggregation layer to integrate features from multiple clients, instead of performing one-to-one feature conversion?
2. Given the FL setting with data heterogeneity, as discussed in the paper, gradient convergence can be problematic for IN training, especially under severe non-IID conditions. It would be better to provide results and discuss in such extreme heterogeneous setting, like iNaturalist dataset [1].
3. Table 6 shows adding noise to 0.8σ will improve performance compared to using no noise. I am interested to know the reason. Is it related to added noise helps mitigate gradient divergence?

*[1] Lai, Fan, et al. "Fedscale: Benchmarking model and system performance of federated learning at scale." International conference on machine learning. PMLR, 2022.*

---

> ### Author Response · Authors · 2025-05-22
> **Response to Reviewer r8oD**
>
> Thank you for your valuable and constructive review. We now address your main concerns as follows, and the updated parts are highlighted in blue.
>
> **W1.** The motivation of this paper is somewhat unclear to me.
>
> **Answer:** Thank you for your valuable comments. The IN features are more like anchors rather than direct labels for the local data features. This design ensures that client models maintain their generalization capability without becoming overly biased toward local datasets.
>
> **W2.** It would be better to provide results and discuss in such extreme heterogeneous setting, like iNaturalist dataset.
>
> **Answer:** Thank you for your valuable comments. Due to the limitations of our machines, we are unable to extend our experiments with additional clients and larger datasets. We plan to conduct these more extensive experiments for FedIN using cloud resources in future work.
>
> **W3.** Table 6 shows adding noise to 0.8σ will improve performance compared to using no noise. I am interested to know the reason. Is it related to added noise helps mitigate gradient divergence?
>
> **Answer:** Yes. We believe this is related to the overfitting problem, which stems from gradient divergence during training. With a slight noise level, the model becomes more robust and less biased toward local datasets, indicating improved generalization capability for clients.

---

### Review · Reviewer_ywu6 · 2025-04-25

**Summary Of Contributions:**

This paper addresses the challenge of system heterogeneity as well as data heterogeneity without requiring any public data in a federated learning setting. Intermediate layers are aligned through a batch of features, provided that the layer shapes match. To manage the connection between local gradient updates and intermediate feature alignment, FedIN incorporates a convex optimization objective.

**Audience:**

Yes

**Claims And Evidence:**

Yes

**Requested Changes:**

1. Please include limitations as a separate section (before the conclusion) and elaborate more on your proposed method of projecting layer dimensions to the same size.

2. Include ViT results in the main paper as they are crucial to support the method's performance claims.

**Strengths And Weaknesses:**

Strengths

1. Authors have greatly improved the paper as compared to their previous submission, and included results on different models like ViTs as suggested.
2. Authors also provide an analysis of privacy by including Gaussian noise in model updates.
3. The proposed technique is described well and is generally easy to follow.
4. Additional baselines are included, which establish the efficacy of this approach.

Weaknesses

1. FedIN would not work with models of different widths because it requires feature alignment between corresponding layers across clients.

---

> ### Author Response · Authors · 2025-05-22
> **Response to Reviewer ywu6**
>
> Thank you for your valuable and constructive review. We now address your main concerns as follows, and the updated parts are highlighted in blue.
>
> **W1.** Please include limitations as a separate section (before the conclusion) and elaborate more on your proposed method of projecting layer dimensions to the same size.
>
> **Answer:** Thank you for your valuable comments. For FedIN, we can deploy layers in the model similar to the early exits mentioned in ScaleFL to serve as the projected layers in FedIN. We have updated this elaboration in the paper.
>
> **W2.** Include ViT results in the main paper as they are crucial to support the method's performance claims.
>
> **Answer:** Thank you for your valuable comments. We have revised this problem in the paper.

---

### Decision · Action_Editor_VLhd · 2025-06-06

**Recommendation:** Reject

**Comment:**

The author response addressed reviewers' concerns on additional communication and computation cost of the proposed algorithm as well as ablation studies on the IN loss. However, most reviewers still have concerns around the design choices and experiments. It is not clear why the proximal term in (3) is included as part of the proposed method, given that this term itself may not bring improvements in all settings (comparing fedprox and fedavg in Table 1). Moreover, reviewers still think that the datasets used are not diverse or interesting enough (e.g., CINIC-10 is similar to CIFAR10), and improvements (within standard deviation in most cases) lack statistical significance (thus not matching what is claimed in the paper).

**Audience:**

This paper should be of interest to audience in distributed learning and federated learning.

**Claims And Evidence:**

The paper claims that the proposed method outperforms state-of-the-art baselines. However, reviewers are concerned that empirical results (e.g., Table 1) do not show clear improvements as the difference can be smaller than standard deviation. Also hyperparameter tuning process is not reported.

**Resubmission Of Major Revision:**

The authors may consider submitting a major revision at a later time.